# Broadband Vibration-Based Energy Harvesting for Wireless Sensor Applications Using Frequency Upconversion

**DOI:** 10.3390/s23115296

**Published:** 2023-06-02

**Authors:** Jinglun Li, Habilou Ouro-Koura, Hannah Arnow, Arian Nowbahari, Matthew Galarza, Meg Obispo, Xing Tong, Mehdi Azadmehr, Einar Halvorsen, Mona M. Hella, John A. Tichy, Diana-Andra Borca-Tasciuc

**Affiliations:** 1Department of Mechanical, Aerospace, and Nuclear Engineering, Rensselaer Polytechnic Institute, Troy, NY 12180, USA; jinglunduran@gmail.com (J.L.); galarm2@rpi.edu (M.G.); obispm@rpi.edu (M.O.); tichyj@rpi.edu (J.A.T.); 2Department of Microsystems, University of South-Eastern Norway, 3184 Borre, Norway; arian.nowbahari@usn.no (A.N.); mehdi.azadmehr@usn.no (M.A.);; 3Department of Electrical, Computer and System Engineering, Rensselaer Polytechnic Institute, Troy, NY 12180, USA; tongx@rpi.edu (X.T.); hellam@rpi.edu (M.M.H.)

**Keywords:** wireless sensor networks, vibrations, energy harvesting, energy conversion, sensors, frequency up-conversion, low-frequency application, high bandwidth, design optimization, experimental testing

## Abstract

Silicon-based kinetic energy converters employing variable capacitors, also known as electrostatic vibration energy harvesters, hold promise as power sources for Internet of Things devices. However, for most wireless applications, such as wearable technology or environmental and structural monitoring, the ambient vibration is often at relatively low frequencies (1–100 Hz). Since the power output of electrostatic harvesters is positively correlated to the frequency of capacitance oscillation, typical electrostatic energy harvesters, designed to match the natural frequency of ambient vibrations, do not produce sufficient power output. Moreover, energy conversion is limited to a narrow range of input frequencies. To address these shortcomings, an impacted-based electrostatic energy harvester is explored experimentally. The impact refers to electrode collision and it triggers frequency upconversion, namely a secondary high-frequency free oscillation of the electrodes overlapping with primary device oscillation tuned to input vibration frequency. The main purpose of high-frequency oscillation is to enable additional energy conversion cycles since this will increase the energy output. The devices investigated were fabricated using a commercial microfabrication foundry process and were experimentally studied. These devices exhibit non-uniform cross-section electrodes and a springless mass. The non-uniform width electrodes were used to prevent pull-in following electrode collision. Springless masses from different materials and sizes, such as 0.5 mm diameter Tungsten carbide, 0.8 mm diameter Tungsten carbide, zirconium dioxide, and silicon nitride, were added in an attempt to force collisions over a range of applied frequencies that would not otherwise result in collisions. The results show that the system operates over a relatively wide frequency range (up to 700 Hz frequency range), with the lower limit far below the natural frequency of the device. The addition of the springless mass successfully increased the device bandwidth. For example, at a low peak-to-peak vibration acceleration of 0.5 g (peak-to-peak), the addition of a zirconium dioxide ball doubled the device’s bandwidth. Testing with different balls indicates that the different sizes and material properties have different effects on the device’s performance, altering its mechanical and electrical damping.

## 1. Introduction

Self-driving cars, smart homes, remote health monitoring, and industrial automation have all made notable advancements and are becoming widespread [1,2,3,4,5,6,7,8,9,10]. In all these applications, which are a part of the Internet of Things (IoT), a variety of embedded and interconnected systems are continuously monitored by wireless sensors [11,12]. The fast expansion of these wireless, low-power IoT devices is propelling the development of microscale energy harvesting solutions. Currently, batteries are the main power sources for wireless devices, which results in extra maintenance costs due to their short lifespan [1,13,14]. Hazardous waste is also a problem given the sheer number of batteries used now to power sensors. Energy harvesting [12,15,16], which describes the technology of scavenging energy from the surrounding sources to power gadgets, is a solution to this issue. Energy harvesting eliminates the need for maintenance and permits continuous power generation from ambient sources [17,18].

Kinetic energy has traditionally been scavenged to power wireless sensor nodes. It is clean, ubiquitous, and inexpensive in comparison to other types of ambient energy. It can be available as a steady source, usually at low frequencies [19,20]. The three main transduction methods used by kinetic energy harvesters are electrostatic [21], piezoelectric [22,23,24], and electromagnetic [25,26,27,28]. However, on the one hand, electromagnetic energy transduction mechanisms can be bulky and hard to miniaturize, and piezoelectric transduction necessitates special piezoelectric materials. On the other hand, due to their compatibility with the silicon-based technologies used for microelectromechanical systems and integrated circuits fabrication [29], and their potential for on-chip integration with power conditioning and sensing circuitry, electrostatic vibration energy harvesters (e-VEHs) remain the preferred energy technology for silicon IoT devices. Still, unlike piezoelectric and electromagnetic harvesters, they are not commercialized yet because of their low power output and narrow frequency range but remain under active research.

A typical MEMS e-VEH transducer consists of a spring-supported shuttle mass that moves with the vibrations, having a set of electrodes on its edges. The moving electrode set is intercalated with a set of fixed electrodes anchored to the substrate, forming a variable capacitor [30]. An ideal e-VEH device should harvest low-frequency vibrations (abundant in the environment) over a wide bandwidth. For an e-VEH to efficiently harvest low-frequency vibrations, its natural frequency must match that of the input mechanical vibration. Operating at natural frequency ensures the maximum amount of energy being converted from the mechanical to the electrical domain. The converted power, however, is proportional to frequency, and despite resonance conditions, the output power of the current e-VEH is insufficient, even for low-duty cycle wireless sensors [19].

There are various strategies explored to increase the output power of e-VEHs. They include geometry optimization [31], combining power from parallel connected devices [32], and frequency upconversion [33]. The first approach, geometry optimization, has been explored in several works [21,31,34]. The supporting spring, electrode shape, and/or shuttle mass were optimized. Given the microscale dimensions of these devices and low-frequency operation, the reported output power remains modest, even when the shape is optimized. Combining several devices in parallel (and operating them under the right conditions) is another approach to increasing the power. Li et al. showed a system consisting of two different e-VEHs connected in parallel that produces higher power over an extended frequency range as compared with a single device [32]. The drawback of this technique is the necessity to employ multiple devices.

Finally, the frequency upconversion involves producing high-frequency mechanical vibrations in response to a low-frequency excitation, usually by triggering a free oscillation due to the impact of the electrodes [2,33]. The impact is facilitated by large excursions of the mobile electrodes in the presence of soft suspension springs of the shuttle mass. The device is still tuned to resonate with input excitation frequency. However, per each half cycle of the input mechanical vibration, multiple electromechanical transductions occur due to the high-frequency free oscillation of the electrodes.

As a result, the output power, which, as was previously mentioned, positively correlates to the output frequency, is increased. This is because, in practice, when implemented, for example, with a charge-constrained circuit, energy can be harvested every time a voltage peak is detected [35].

A previous study focusing on the electrode impact highlighted the critical importance of slanted electrode sidewalls, showing that a non-uniform electrode gap during impact leads to weaker electrostatic forces and prevents pull-in [33].

Another approach to creating up-converted frequency from impact is based on a springless mass (a ball) fitted inside a slit cut in the shuttle mass. During external vibration, the ball collides randomly with the shuttle mass, triggering high-frequency vibrations, and effectively extracting energy at low frequencies (10–60 Hz) [19,36]. So far, the experimental and theoretical studies exploring this technique all focused on a single material and ball size [37,38]. Even though a heavier ball is believed to be beneficial to the power output, it may lead to structural damage to the device in the long run. At present, there is no understanding of the benefit of having balls of different materials and sizes. Furthermore, the main advantage of employing impact-based frequency upconversion is its design simplicity.

This paper investigates experimentally e-VEH devices with frequency upconversion due to the impact. The device design combines a non-uniform electrode design and a springless mass. As mentioned previously, to prevent pull-in following electrode impacts, a non-uniform electrode gap is necessary [2,33]. In a previous experimental study on impact-based e-VEHs [2], the devices were fabricated in a research cleanroom using an atypical deep reactive ion etching process to produce sloped (non-vertical) electrode walls. In the current study, the devices feature a non-uniform gap coming from the non-uniform electrode width. The electrode walls are designed to be vertical. This design allows for the use of common microfabrication techniques available at commercial foundries for device microfabrication. Specifically, a standard SOIMUMPs method available at MEMSCAP, Inc., was used to fabricate the devices. A springless mass (a microball) was added to a cavity etched in the shuttle mass to investigate the benefit of combining the two impact-based methods for frequency upconversion. The effects of the ball size and material properties on the output voltage and device bandwidth were also studied. The frequency upconversion methods used here, specifically the electrodes’ impact and springless mass impact on the shuttle mass, as well as the nonlinear springs and the addition of soft stoppers, are also expected to increase the bandwidth of the proposed device.

Since it can function over a wide range of vibration frequencies, this device could have a wide range of potential applications from powering medical devices via human motion to powering sensors on bridges, automobiles industrial equipment, or HVAC systems [2,4,12].

Portions of this paper previously appeared as reference [17].

## 2. Device Description

### 2.1. Design Concept

Figure 1a is a simplified schematic of the in-plane gap-closing electrostatic energy harvester investigated here. The device uses relative motion to enable a variation of the capacitance formed between its mobile and fixed electrodes. Under the voltage or charge constraint, this variation of capacitance produces a flow of charge that can be harvested with an appropriate interface electrical circuit to recharge a battery (for the voltage constraint) or increase the potential across a storage capacitor (for the charge constraint) [29,34,39]. The designed device consists of a rectangular shuttle mass suspended by a serpentine spring in each corner. It consists of a rectangular shuttle mass suspended by a serpentine spring in each corner. The movable electrode set (light blue) on the two parallel edges of the shuttle mass is interdigitated with the electrode set (gray) fixed to the substrate, forming a variable capacitor. In the center of the shuttle mass, there is an opening large enough to host a ball of a diameter of order 1 mm, which can move within the cavity in response to the applied external vibration. During vibrations, the distance between the moveable electrodes and the fixed electrodes changes as the shuttle mass travels, causing the capacitance to change. Two sets of soft stoppers (anchored cantilever beams) are placed on each side of the shuttle mass. Their role is to engage with the shuttle mass at a large vibration amplitude and diminish the force experienced when the electrodes collide. A cross-section view of the device is shown in Figure 1b, detailing the materials used in each layer of the device.

For testing, the harvester is mounted on a printed circuit board (PCB) as shown in Figure 2a, and protected with a clear poly(methyl methacrylate) (PMMA) cover to keep dust off. A picture of the serpentine spring supporting the shuttle mass is shown in Figure 2b.

Figure 2c shows the electrode set. The widths of the electrodes are non-uniform. Two adjacent electrodes have asymmetric width variations. Specifically, the electrodes attached to the shuttle mass have wide bases and narrow tips, while the electrodes attached to the anchor have wide tips and narrow bases. This geometry results in an electrode gap of a trapezoidal shape as shown in Figure 2c. This gap ensures separation even when the electrodes touch, as shown in Figure 2d, decreasing the risk of pull-in effects. The electrodes are covered by a 200 nm thick layer of parylene C, which serves as an electrical insulator and prevents shorting when the electrodes collide. Figure 3 displays the scanning electron microscope (SEM) pictures of the electrodes covered with parylene, springs, stoppers, and wire bonding on a bonding pad.

The e-VEH was fabricated using the silicon-on-insulator (SOI) process at MEMSCAP. Inc. (Durham, NC, USA), following the foundry’s design guidelines detailed in [40]. While the SOIMUMPs process at MEMSCAP used to manufacture these devices greatly restricts device power because of strict limitations on the device size and geometry, it was preferred over custom fabrication due to its high yield. Despite the low power output, MEMSCAP-fabricated devices can still be used for the proof of concept of the proposed approach and investigating ball effects. Furthermore, the performance of the device may be qualitatively compared to a device with a similar size, as previously reported in the literature [1,32].

Table 1 lists all relevant device dimensions. The device layer is 25 μm thick. The oxide layer is 2 μm thick and the handle silicon layer that supports the device is 400 µm thick. The handle silicon and oxide layers are stripped away from the mobile components of the device. The SOIMUMPs process at MEMSCAP limits the layer thickness (up to 25 μm) and restricts the die size to 11.15 mm × 11.15 mm.

### 2.2. Device Fabrication and Packaging Process

The microfabrication process was performed in two batches with different yields. According to the manufacturer, the first batch resulted in a 20% yield while the second batch resulted in an 80% yield with an improved process. The first batch used the standard process in [40], whereas for the second batch, stretching of the silicon dies was avoided by not using a thermal tape to fix the wafer to the substrate during the laser dicing which improved the dicing process’ yield. The etched silicon dies were shipped in appropriate containers. Once received, the silicon dies were first removed from their containers inside a ’cleanroom’ and inspected under an optical microscope for detecting visible defects. Then, each silicon die underwent parylene C deposition using a parylene coater (Union Carbide 1050). The thickness of the parylene layer was characterized after deposition using a profilometer and it was found to be approximately 0.2 μm. Next, the silicon die was wire-bonded (using a West Bond machine) to a custom-designed PCB for electrical characterization and testing. The metal bonding pads on the silicon die were designed to reduce parasitic capacitance. In addition, a grounding pad was placed on the silicon oxide layer for the same purpose. The final assembly step consisted of placing a protective acrylic glass cover on top of the silicon die and PCB. The acrylic glass cover was removable and could be detached to place a microball. The microball rested on a copper surface that was connected to the electrical ground.

## 3. Experimental Setup

The experimental configuration employed for the device characterization is depicted in Figure 4a. This setup and the electrical circuit depicted in Figure 4b are commonly used for assessing the energy conversion capability of such variable capacitors [2]. Note that the testing circuit is distinct from interface circuits required to integrate with the device for creating a micro-power generator.

The PCB supporting the device was installed on a shaker table (Labworks ET-126-4). A sinusoidal wave signal with a specific frequency was produced by a function generator (Agilent 33210A) and amplified with a signal amplifier (Piezo Systems, Model EPA-104). The amplitude and frequency of the mechanical vibration were generated by the shaker, which was fed by the signal amplifier. The excitation amplitude and frequency were measured by an accelerometer (Piezotronics 333B50) mounted on the shaker table. The devices were tested with mechanical vibration frequencies ranging from 5 to 1000 Hz.

The input port of e-VEH was connected to a DC bias voltage supply (Agilent E3631A), whereas the output port was connected to a load resistor used for electrical characterization, such as voltage and power. A NI Data Acquisition (NI-9215) system connected to LabVIEW software was used to measure the root mean square (RMS) output voltage across the load resistor, and the mechanical excitation’s acceleration amplitude and frequency. The voltage across the resistor was also observed using an oscilloscope (Keysight InfiniiVision DSOX2004A) for viewing the signal and capturing the free oscillations triggered by impact. The circuit shown in Figure 4b was used to measure the output voltage across the series resistance *R_load_* during mechanical oscillations. The internal load resistance of the oscilloscope was 1 MΩ, which means the load was dominated by the measurement apparatus and not by the 72 MΩ resistor.

## 4. Results

### 4.1. Device Response without a Microball

The main objective of this paper is to understand changes in the performance of a device designed to exhibit inherent electrode impacts when microballs of different materials and sizes are being incorporated in the device shuttle mass, triggering additional impact. To better understand the effect of the springless mass on performance, the devices were first characterized without a microball.

The first step in testing was to examine all the devices without any noticeable defects and understand the repeatability of their responses to the same input signal. As presented in [41], electrostatic vibration energy harvesters, such as the one designed here, are highly nonlinear devices with high sensitivity to slight changes in device parameters, which may occur due to typical variations associated with the microfabrication process.

This examination was done by performing a frequency sweep over the same frequency range of applied vibration while keeping the vibration amplitude at 1 g, where g is the gravitational constant and the same applied bias voltage at 3 V. The resistance in series with these devices was the same, at 72 MΩ. The results for all devices are shown in Figure 5. The results illustrated in Figure 5 are for an upsweep frequency scan. Due to the nonlinearity of these devices and the existence of different orbits of oscillations, depending on initial conditions, a different response is expected for a downsweep frequency scan as seen in the measurements presented later in this section.

As seen in Figure 5, out of the eleven devices tested, there seem to be two categories of devices having narrower (80–280 Hz) and wider bandwidths (80–380 Hz), respectively. A plausible explanation for this difference in frequency response is that the devices, which resulted from two different batches, may have been subjected to different etching undercuts, or slight differences may have occurred during the multiple steps throughout the SOIMUMPs process and accumulated. Very small differences in the etching rates and etch undercuts across the wafer surface area could be the principal cause. Despite the great progress made in the industry regarding control over this aspect, the thicknesses of these devices are at the upper limits of the foundry design rules, where this particular fabrication step could be somewhat compromised. Another possible explanation for the differences seen in Figure 5 could be the non-uniform intrinsic stress of the device layer [1], which could lead to a non-uniform suspension beam and shuttle mass upward curvature. Such differences may have had an effect on the stiffness of the spring and end stoppers, as well as the area contributing to electrostatic and squeeze film-damping forces, thereby influencing the behavior of the devices. While it is out of the scope of this paper to explore in-depth the reasons causing the differences seen in device response, this issue was brought up here to highlight device sensitivity to the fabrication process. Such issues become critically important if these devices are to be manufactured on a large scale for commercialization, yet they are scarcely discussed in the literature. It is important to note that small temperature and humidity variations inside the lab had minimal impact on the output of the device under the same vibration conditions.

For all subsequent studies presented in this paper, the devices with the wider bandwidths were used in testing. Furthermore, to assess the reliability of the devices, they underwent continuous testing for three months and multiple data collection sessions were conducted.

Figure 6 shows a comparison of the measured RMS output voltage between the device under consideration with a die size of 11.15 mm × 11.15 mm and another device from [32] with a similar size (11 mm × 11 mm), with similar device thicknesses, and made using the MEMSCAP SOIMUMPs fabrication process. The range of frequencies where the power is 0.707 of the peak power is known as the half-power or 3-dB bandwidth, indicated by a blue arrow. This figure compares the two techniques to broaden the bandwidth: Figure 6a utilizes a parallel system with two similar devices, while Figure 6b employs frequency upconversion with a single device. The measurement is performed without a ball under 1 g peak-to-peak acceleration, 3 V bias voltage, and different ranges for input vibration frequency centered around each device’s natural frequency. As can be seen, this new device design featuring non-uniform cross-section electrodes (Figure 6b) exhibits seven times the bandwidth over the parallel device system (Figure 6a). However, the strong non-linearity due to the electrodes’ shape results in a hysteresis, which only appears at higher voltages and acceleration testing conditions for the parallel device system, where the interdigitated electrodes have a uniform gap [32].

Figure 7 displays the frequency sweeps for device no. 1 (as labeled in Figure 5), showing the RMS output voltage for upsweeps and downsweeps at three different excitation accelerations (1 g, 1.5 g, and 2 g) with a bias voltage of 4 V. As seen in previous literature studies [1], the device exhibits the hysteresis typical to this electromechanical system. Higher vibration amplitudes, where the shuttle mass displacement is greater, are likely to have nonlinear effects. The up- and down-frequency sweeps were performed with increasing vibration amplitudes to illustrate this behavior.

Figure 7 also shows that there is an increase in the device’s RMS output voltage and frequency response range with an increasing acceleration amplitude, as previously reported [1]. Bandwidth broadening and sweep direction hysteresis with jump-up and jump-down phenomena manifest if a threshold acceleration is exceeded. These characteristics are due to spring hardening, where the peak response is at a frequency below the natural resonance of the microstructure and the frequency response curve is slanted towards higher frequencies.

Figure 8 displays a frequency sweep example for device no. 1, showing the RMS output voltage for upsweeps and downsweeps at three different bias voltages (2 V, 4 V, and 6 V) with an input peak-to-peak acceleration of 1 g. For this scenario, the output voltage increases almost linearly with a larger bias voltage; however, the frequency response of the device is not affected much by varying the bias voltage. The hysteresis is still present for similar reasons as mentioned before.

While the trends discussed so far are typical to e-VEHs with or without impacts [1,33], what is special about these devices is the ringing exhibited following electrode collision, which produces frequency upconversion. Frequency upconversion is a beneficial effect as energy can, in principle, be harvested with the appropriate circuit any time there is a peak in voltage [14,37]. While frequency upconversion due to the electrode impact has been reported previously, it was done by employing slanted electrode walls. The non-uniform gap between the slanted electrode wall was found to be critical in preventing pull-in during collision [38]. Since slanted walls require an atypical etch, the electrode geometry investigated here is considered instead a non-uniform cross-section electrode. In this way, the gap between electrodes is still non-uniform, while the electrode wall is vertical, which can be obtained with a standard deep etch. Figure 9 shows an example of the ringing phenomenon obtained with device no. 6, without a microball. The device was tested under a bias DC voltage of 3 V and an external vibration frequency of 17 Hz with 2 g peak-to-peak acceleration. The waveform of the peak-to-peak acceleration is shown by the sinusoidal (orange) signal. The device output is the blue signal. The many peaks that are congregated are common in devices that experience electrode impacts and are brought on by secondary oscillations (free vibration) of the electrodes that occur because of an impact. There are some differences between the two consecutive groups of peaks, which are most likely the result of the two fixed electrodes on either side of a movable finger electrode oscillating out of phase. The sequence, however, is repeated and stable over time. These results show that a successful impact, avoiding pull-in, can be obtained with this new electrode geometry. It is important to note that this frequency upconversion (ringing) is obtained at a low vibration frequency of 17 Hz, which is within the range of low-frequency vibrations from common ambient sources. Therefore, this new design is beneficial for most practical applications.

### 4.2. Device Behavior with Microballs

The tests were repeated, and the results were collected, as described in the previous section, with a microball added to the shuttle mass cavity. The microballs used for testing are listed in Table 2, along with their distinctive parameters. All the balls used have a smooth finish. Despite having a similar diameter size of 0.8 mm, the zirconium dioxide is denser than the silicon nitride microball and, thus, has more mass. Furthermore, with a small diameter of 0.5 mm, the Tungsten carbide ball has more mass compared to the larger-diameter silicon nitride ball. While mass is an important parameter and is expected to influence the results, the microball’s electrical charging was found to be an issue. Specifically, a larger selection of microballs was initially attempted, but several material types, such as aluminum and chrome, did not yield successful results; the microball did not move effectively under external vibrations, leading to device failure.

Figure 10 shows an example of electrode ringing obtained with this new electrode design, with two different microballs: (a) Tungsten carbide microball of 0.5 mm diameter; (b) Tungsten carbide microball of 0.8 mm diameter. The testing condition was 1.5 g, 15 Hz, at 3 V DC bias voltage. The plots depict the RMS output voltage of device no. 6 as a function of time. The collision between the shuttle mass and the microball adds additional ringing peaks as compared with the heavier microball in Figure 9. Specifically, compared to Figure 10a, Figure 10b shows that the larger Tungsten microball (0.8 mm) increased the number of peaks, as well as their amplitude, which translates into more power harvesting potential. However, the 0.8 mm Tungsten carbide microball was found to increase the risk of damaging the silicon dies due to its heavy mass.

For all subsequent tests, device no. 5 was used. Due to the frequent handling of a device when adding and removing a microball, several devices were lost during microball testing, including device no. 1, which was used to study the device response without a microball. However, as shown in Figure 5, devices no. 1 and no. 5 have very similar behavior and a comparison between tests with and without the microball is meaningful.

The results shown in Figure 11, Figure 12, Figure 13, Figure 14, Figure 15 and Figure 16 illustrate the RMS output voltage as a function of frequency for upsweep and downsweep scans with and without a microball placed inside the cavity with different peak-to-peak acceleration levels and bias voltages.

Figure 11 shows the graph of the RMS output voltage as a function of frequency (upsweep and downsweep) with a constant bias voltage of 1.5 V and a peak-to-peak acceleration amplitude of 0.5 g for mechanical vibration. Inspecting the bandwidth, for the upsweep frequency tests, the bandwidth of the device without a microball (black) is lower compared to when the microballs are added. The 0.8 mm zirconium dioxide ball showed the highest device response bandwidth (green), which is more than double that in the case when there is no ball. The Tungsten (red) and silicon nitride (blue) balls show similar responses. It can be concluded that a possible explanation for this increase is that the collisions of the heavier zirconium dioxide ball with the shuttle mass result in the impact of the shuttle mass with the soft stoppers and electrodes, thus changing the effective spring constant of the device, as discussed in [41]. The short bandwidth of the device without a ball may be explained by the absence of a collision with the end stoppers and the electrodes at the low excitation peak-to-peak acceleration.

For all tests, the experimental downsweep results show almost a similar trend but with a narrower bandwidth due to the hysteresis in the spring response, as discussed previously.

With the same bias voltage of 1.5 V, and an increased peak-to-peak acceleration of 1 g, Figure 12 demonstrates that all the bandwidths during testing have increased compared to Figure 11, for frequency upsweep tests. Similar to Figure 11, the zirconia (green), silicon nitride (blue), and Tungsten balls (red) exhibited voltage outputs at low vibration frequencies, between 5 and 90 Hz, whereas the device without the ball (blue) did not show any output. This behavior was witnessed for all test results when the microballs were added compared to when there was no microball. The device without a ball exhibits a similar bandwidth as when the zirconia ball was added. However, the RMS voltage is higher for the latter. Similar to the previous case with a peak-to-peak acceleration of 0.5 g, the Tungsten and silicon nitride balls show similar bandwidths, but their bandwidths and RMS voltage amplitudes are lower compared to the no-ball and zirconia cases. The impact of the shuttle mass on the soft stoppers and the electrodes at 1 g increased the excitation bandwidth of the device from 90 Hz upwards, even if no ball was added.

Figure 13 shows the results for tests conducted at a bias voltage of 1.5 V, similar to Figure 11 and Figure 12, but with an increased peak-to-peak acceleration amplitude of 2 g. The zirconia ball test shows the largest bandwidth, up to 700 Hz. Unlike the device without the ball, adding a microball clearly has an effect on device output at low frequencies (<50 Hz), producing a voltage of order 0.5 mV, a trend similar to the one previously reported in the literature [42]. Thus, it appears that adding the microballs is then beneficial in practical situations, where most vibration sources are at low frequencies and there is a sufficient peak-to-peak acceleration amplitude. Similar to previous tests, the silicon nitride and Tungsten balls have similar bandwidths, which are slightly lower than the device without the ball.

The next series of tests were performed with a bias voltage of 3 V and similar peak-to-peak amplitude accelerations, as seen in Figure 11, Figure 12 and Figure 13, namely, 0.5 g, 1 g, and 2 g, respectively. Figure 14 shows the results of the device tested under a peak-to-peak acceleration of 0.5 g. Compared with the results shown in Figure 11, the device bandwidth with different ball sizes shows similar trends, namely the heaviest ball, zirconia, has the largest effect on the bandwidth, while the other two balls of similar mass produce similar bandwidth increases. In all tests, the RMS output voltage amplitude increased due to the increase of the bias voltage. However, doubling the bias voltage from 1.5 V to 3 V had a minor effect on the device’s response bandwidth at this low acceleration.

With a bias voltage of 3 V and 1 g peak-to-peak acceleration, the test results shown in Figure 15 reveal a wider device bandwidth when no microball is added at vibration frequencies higher than 90 Hz. All tests with all microballs have similar trends and comparable bandwidths. Compared with Figure 11, the zirconia ball test shows a slight decrease in the bandwidth, which may be due to the increased contribution of the electrical damping caused by the high bias voltage.

Figure 16 shows similar bandwidths at higher vibration frequencies (>90 Hz) except for the test with the 0.5 mm Tungsten microball, which exhibits a drastically reduced bandwidth. The silicon nitride ball gives the highest RMS output voltage among the ball testing but is slightly lower compared with the no-ball testing. Compared with Figure 13, the bandwidth from the zirconia ball test has decreased by approximately 100 Hz, while for the Tungsten ball, it has decreased by 175 Hz; the silicon nitride ball shows only a slight increase in bandwidth. The device with no ball shows similar bandwidth to that shown in Figure 13. The added bias voltage and the chaotic motion of the microballs, coupled with the changes in electrical damping, may explain these results. The mechanical and electrical properties of the different microballs, such as the friction coefficient and a surface charge, may also affect the device’s damping and, thus, its bandwidth.

In summary, with or without a microball, the RMS output voltage increases strongly when the bias voltage is increased from 1.5 V (Figure 11, Figure 12 and Figure 13) to 3 V (Figure 14, Figure 15 and Figure 16), depending on the applied peak-to-peak acceleration. Specifically, at 0.5 g and 1 g, the RMS output voltage increases by up to 300% when doubling the bias voltage, depending on the ball materials and sizes, while at 2 g, it increases by approximately 200%. Increasing the excitation peak-to-peak acceleration amplitude from 0.5 g to 2 g at a fixed bias voltage produced a less pronounced increase in voltage but a significant change in the range of frequency of the device response. On average, the device frequency response more than doubled when the peak-to-peak acceleration amplitude was increased from 0.5 g to 2 g, for all cases considered.

As stated previously, all these observations are true for the devices with and without a ball. Inspecting the same figures, one can see that differences between the responses with and without a microball depend strongly on the operation parameters. For instance, focusing on a frequency upsweep scan, at a low acceleration amplitude (0.5 g), adding a ball clearly benefits the RMS output voltage for both bias voltages tested. The device with the highest mass ball (zirconium dioxide—the green line) has a stronger effect on both the voltage output and range of frequency for the device response. Specifically, tests with this microball show a 50% increase in the device frequency range for operation and up to a 30% increase in the maximum recorded RMS output voltage as compared to the results for the device without a ball. For these operation conditions, the device responds with the other two microballs in approximately the same way. However, when the peak-to-peak acceleration amplitude is increased, these benefits disappear. In fact, at 1.5 and 2 g, the device without the ball outperforms most other tests with the ball, with a single exception, namely tests with zirconium dioxide at a 1.5 V bias voltage and 2 g.

All the trends discussed above focus on upsweep frequency scans. When looking at frequency downsweeps, the differences between the response of the devices with and without a ball are indistinguishable for most cases, including 0.5 g tests.

## 5. Conclusions

This paper reports on the design, fabrication, and characterization of an electrostatic MEMS energy harvester with a non-uniform comb design to explore its potential for frequency upconversion. A microball added within a cavity of the shuttle mass, which serves as an additional springless mass to increase inertia, is also explored as an add-on technique for frequency upconversion. While it has been previously reported that nonuniform electrodes with slopped walls may be used to increase the output frequency and device power, creating linearly sloping sidewalls for such devices requires a unique DRIE technique. In contrast, the novel trapezoidal-shaped electrode design explored here is compatible with common MEMS production techniques. The effort presented in this paper advanced the state of the art in the following ways: (i) non-uniform cross-section electrodes were created to avoid pull-in and produce high-frequency secondary oscillations after electrode impacts; (ii) to the best of the authors’ knowledge, commercial manufacturing was used to achieve frequency upconversion for the first time; (iii) a springless microball was integrated into the state-of-the-art electrode design to further boost the bandwidth and power at low peak-to-peak accelerations.

From the experimental test results, it can be concluded that the device without a microball showed a maximum RMS output voltage of 8.86 mV at a 2 g peak-to-peak vibration acceleration, a frequency of 658.3 Hz, with a 3 V bias voltage. The designed device exhibited a wider bandwidth, seven times the bandwidth of two previously reported devices connected in parallel with similar dimensions. Adding a microball to the device is beneficial to increasing the bandwidth and output voltage at a low peak-to-peak acceleration amplitude (0.5 g), which is of relevance to applications as most environmental vibrations have low peak-to-peak amplitude accelerations. The 0.8 mm zirconium dioxide ball is beneficial to increasing the overall device’s maximum bandwidth at a constant acceleration of 1.5 g. Theoretically, choosing a heavier mass for the microball appears to be more beneficial, but in practice, heavier masses may lead to damage to the silicon die or the overall device structure. Therefore, choosing the right material and size for the microball is crucial for long-term operation and for the specific application of this device. This proof-of-concept study may open new avenues for improving the power output and frequency responses of electrostatic energy harvesters, bringing them one step closer to being used in real applications. This work could be used as a stepping stone to develop a theoretical model to capture the nonlinear behavior of the designed device as well as the impact of the different microballs.

Modeling would provide a more comprehensive understanding of the effect of the different ball sizes and material properties on the mechanical and electrical damping of the device. This will help optimize the device for practical applications. This broadband energy harvesting device could have a wide variety of potential application fields, ranging from medical fields to bridges, HVAC air ducts, and automobiles, due to its large bandwidth.

## Figures and Tables

**Figure 1 sensors-23-05296-f001:**
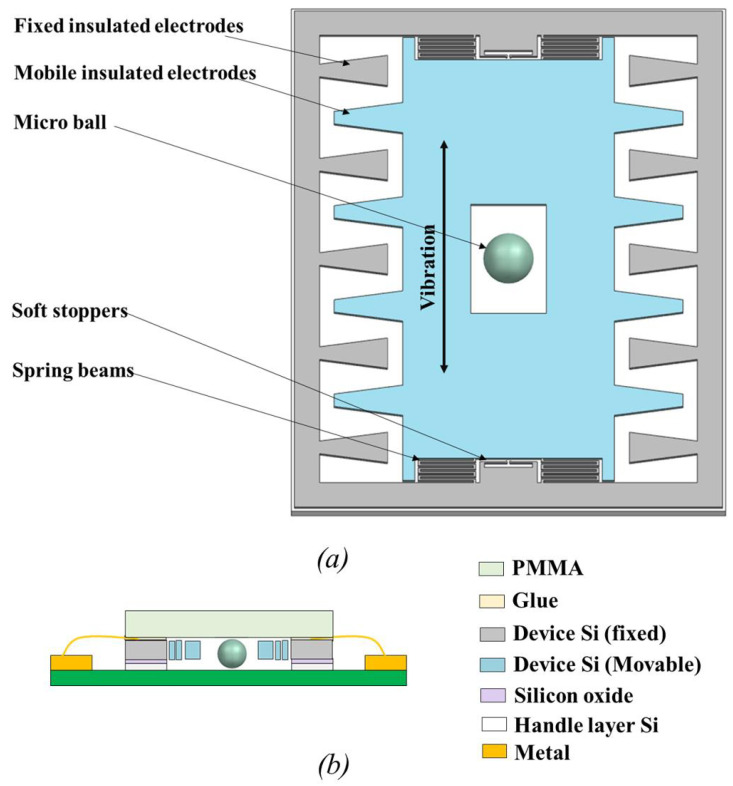
(**a**) A simplified schematic of the e-VEH device, (**b**) cross-section view of the layer composition. ©[2023] IEEE.

**Figure 2 sensors-23-05296-f002:**
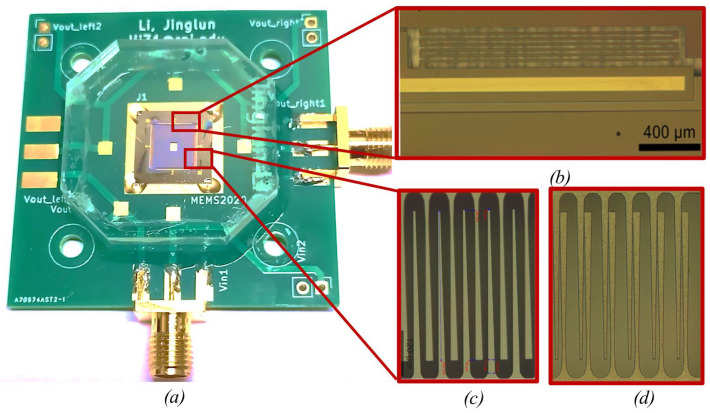
Photographs of the e-VEH: (**a**) The device mounted on a PCB. Close-up images of the structures: (**b**) the suspension spring and the trapezoidal electrodes at (**c**) minimum and (**d**) maximum capacitance positions. ©[2023] IEEE.

**Figure 3 sensors-23-05296-f003:**
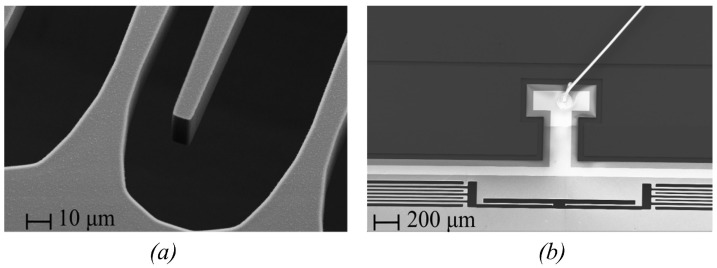
SEM pictures of (**a**) electrodes covered with parylene layer, (**b**) springs, soft stoppers, and wire bonding on the bonding pad.

**Figure 4 sensors-23-05296-f004:**
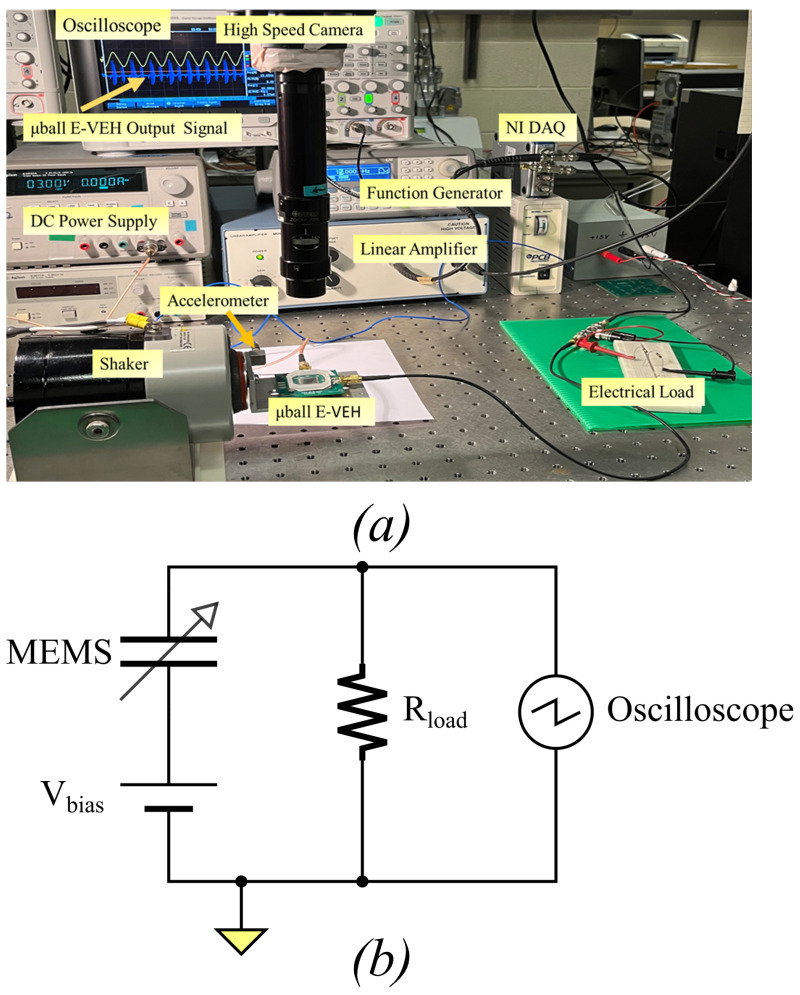
(**a**) Testing setup of the MEMS characterization system ©[2023] IEEE, (**b**) testing circuit schematic.

**Figure 5 sensors-23-05296-f005:**
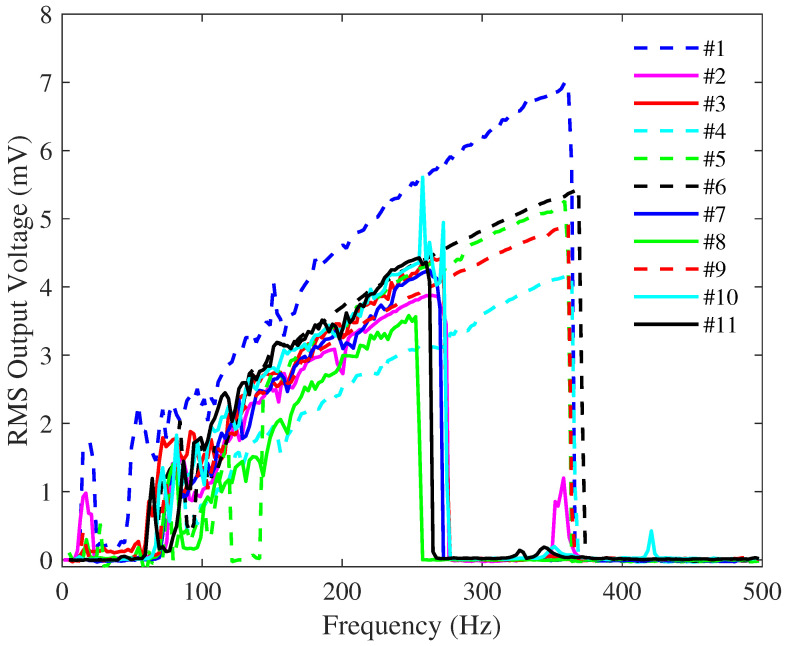
RMS voltage output vs. frequency of applied vibration for all 11 devices without visible defects. In all tests, the vibration peak-to-peak amplitude was 1 g, the bias DC voltage was 3 V, and the resistance was 72 MΩ.

**Figure 6 sensors-23-05296-f006:**
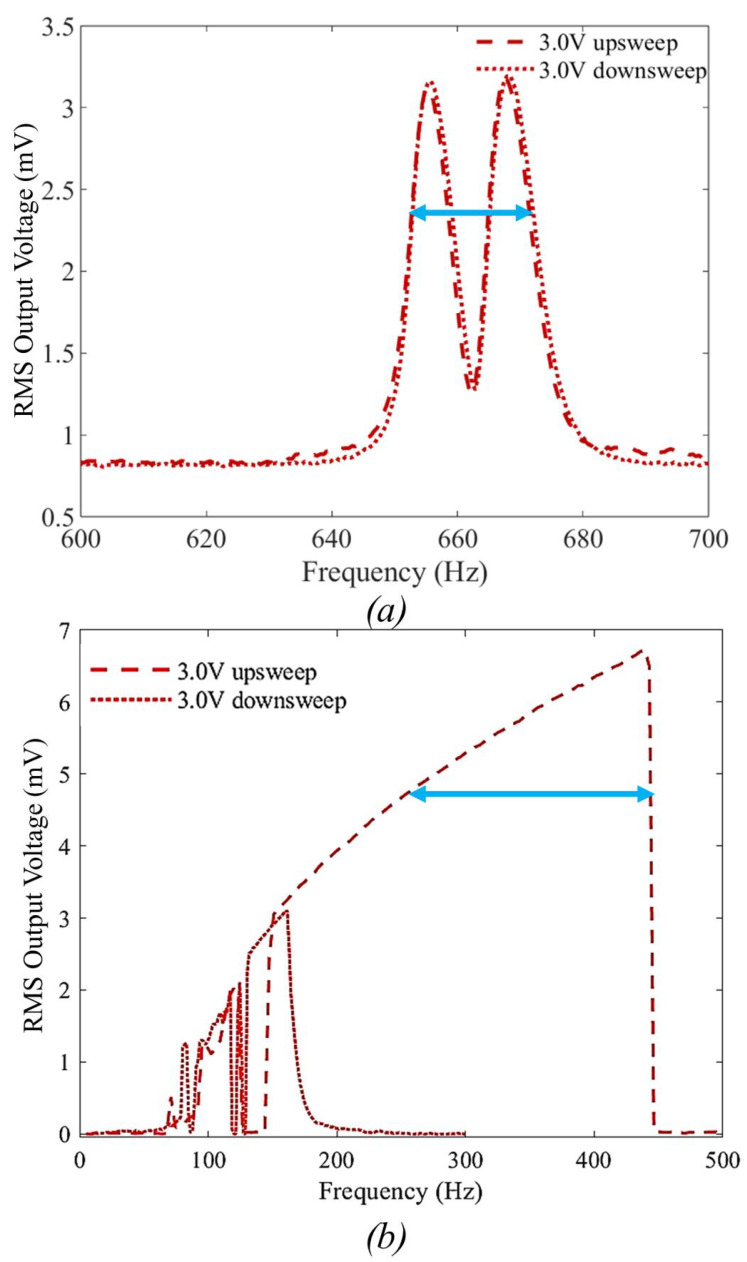
Measured RMS output voltage from the device without a ball as a function of input vibration frequencies under 1 g peak-to-peak acceleration, and 3 V bias voltage from (**a**) a system consisting of two parallel devices with a similar size to the current device [32], (**b**) the current device. The blue line shows the 3 dB bandwidth.

**Figure 7 sensors-23-05296-f007:**
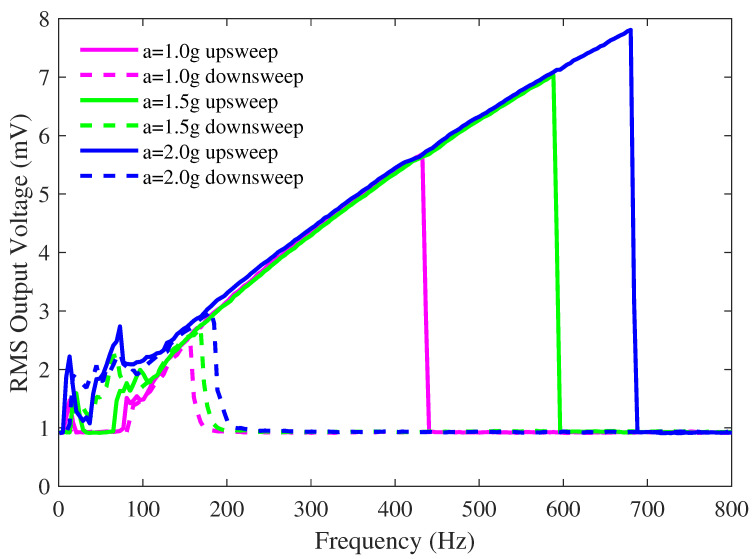
Measured RMS voltage as a function of excitation frequency for a device without a ball as a function of input vibration frequency under different amplitudes of input peak-to-peak acceleration. The bias voltage is 4 V. ©[2023] IEEE.

**Figure 8 sensors-23-05296-f008:**
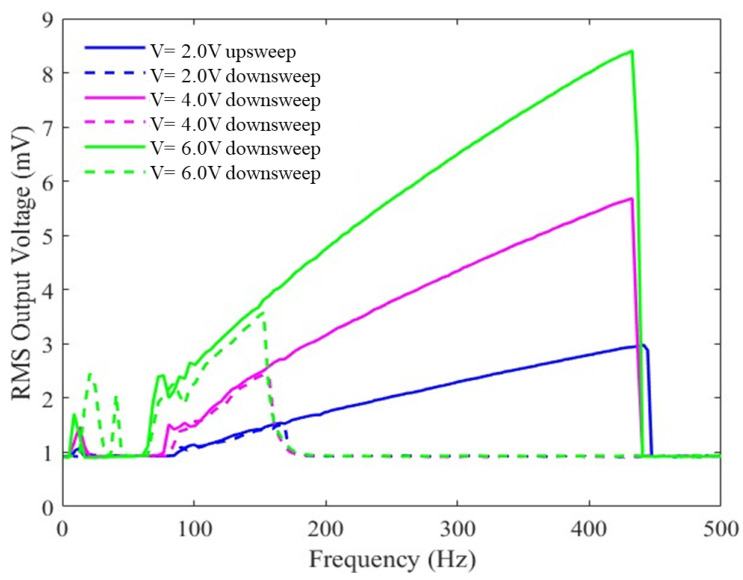
Measured RMS voltage from the device without a ball as a function of the input vibration frequency under different DC bias voltages. The constant amplitude of the input’s peak-to-peak acceleration is 1 g.

**Figure 9 sensors-23-05296-f009:**
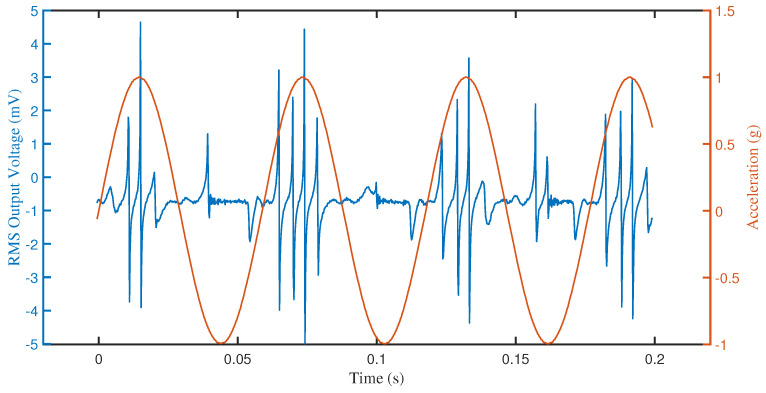
The measured RMS output voltage from device no. 6 without a microball, showing ringing. The test was conducted using a 3 V bias voltage and an input mechanical vibration of 2 g peak-to-peak acceleration at 17 Hz.

**Figure 10 sensors-23-05296-f010:**
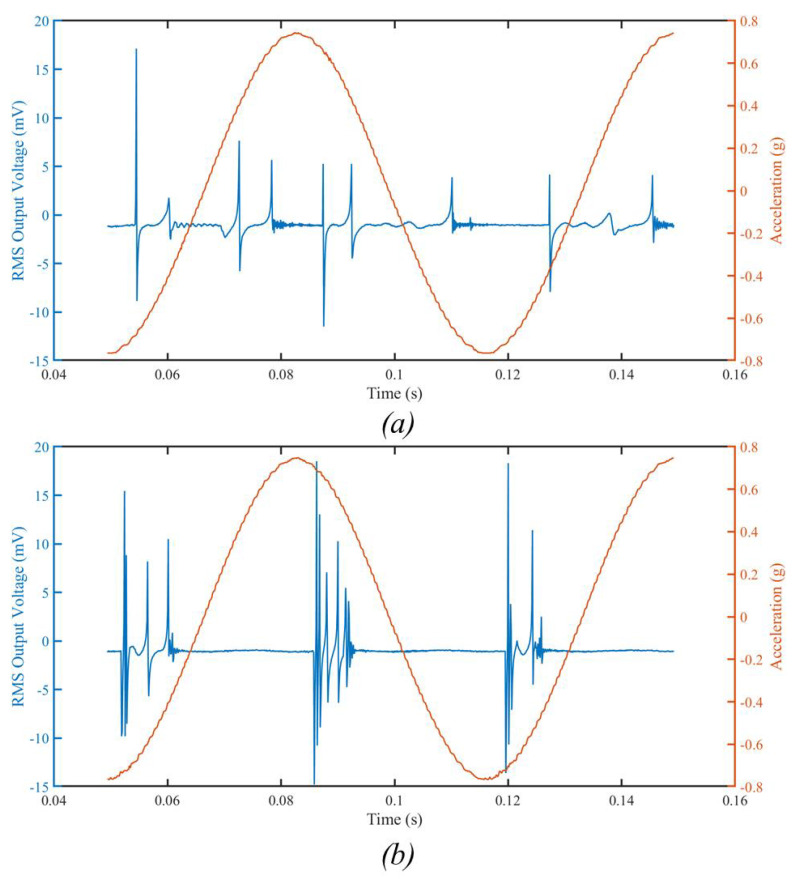
The measured RMS output voltage from device no. 6 showing ringing: 3 V bias voltage and an input mechanical vibration of (**a**) 1.5 g peak-to-peak acceleration, at 15 Hz, with a Tungsten microball that is 0.5 mm in diameter; (**b**) 1.5 g peak-to-peak acceleration, at 15 Hz, with a Tungsten microball that is 0.8 mm in diameter.

**Figure 11 sensors-23-05296-f011:**
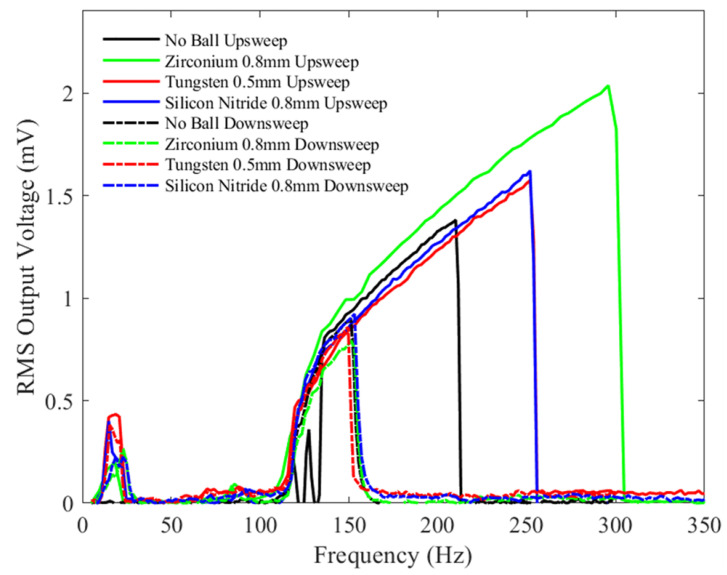
The measured RMS voltage output from the device with different balls under a 1.5 V bias voltage and an input mechanical vibration of 0.5 g peak-to-peak acceleration under a variety of frequencies.

**Figure 12 sensors-23-05296-f012:**
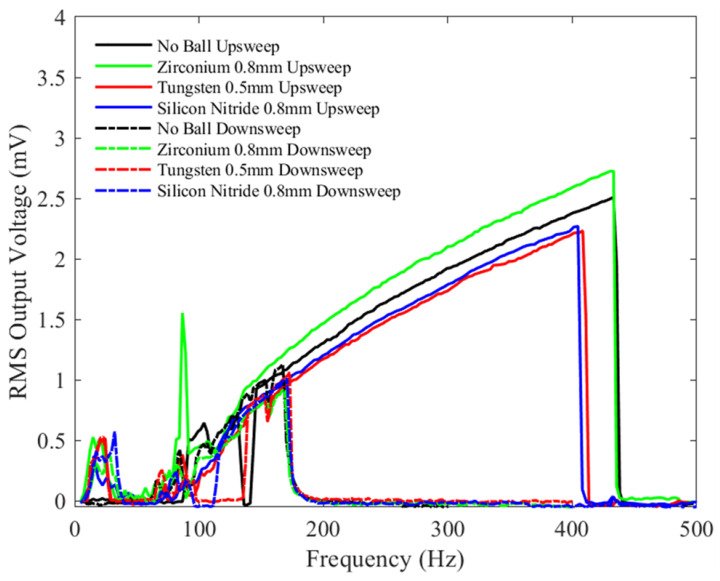
The measured RMS voltage output from the device with different balls under a 1.5 V bias voltage and an input mechanical vibration of 1 g peak-to-peak acceleration under a variety of frequencies.

**Figure 13 sensors-23-05296-f013:**
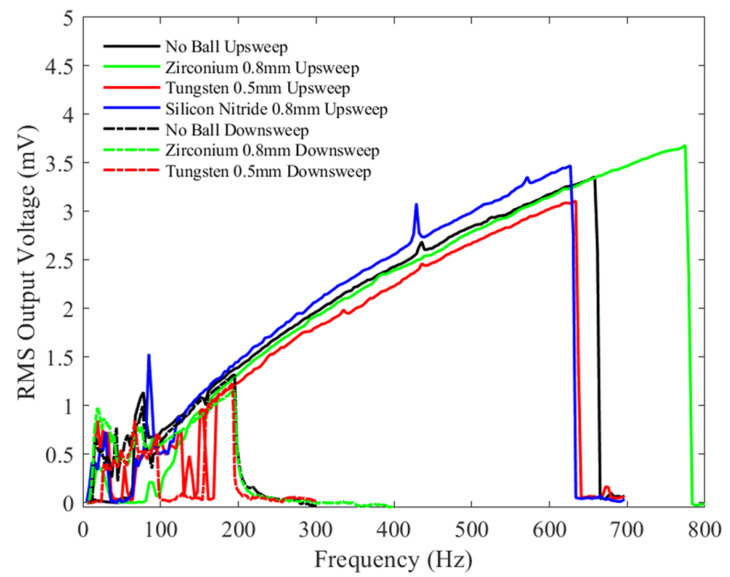
The measured RMS voltage output from the device with different balls under a 1.5 V bias voltage and an input mechanical vibration of 2 g peak-to-peak acceleration under a variety of frequencies.

**Figure 14 sensors-23-05296-f014:**
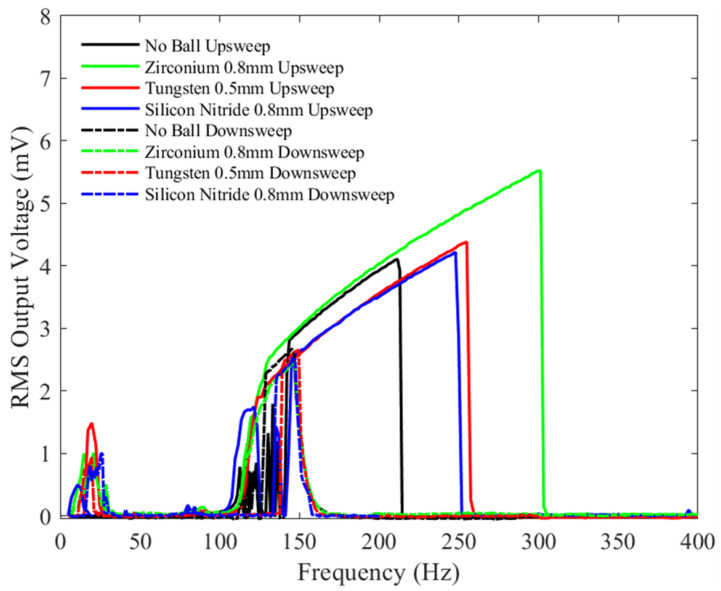
The measured RMS voltage output from the device with different balls under a 3 V bias voltage and an input mechanical vibration of 0.5 g peak-to-peak acceleration under a variety of frequencies.

**Figure 15 sensors-23-05296-f015:**
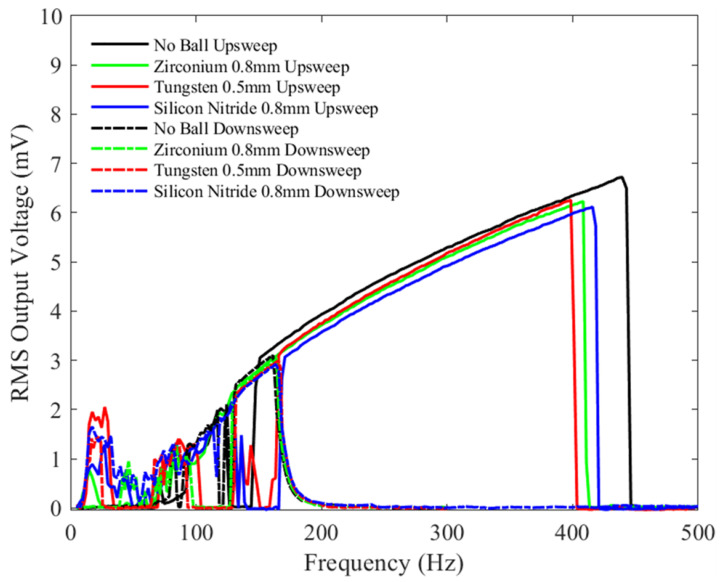
The measured RMS voltage output from the device with different balls under a 3 V bias voltage and an input mechanical vibration of 1 g peak-to-peak acceleration under a variety of frequencies.

**Figure 16 sensors-23-05296-f016:**
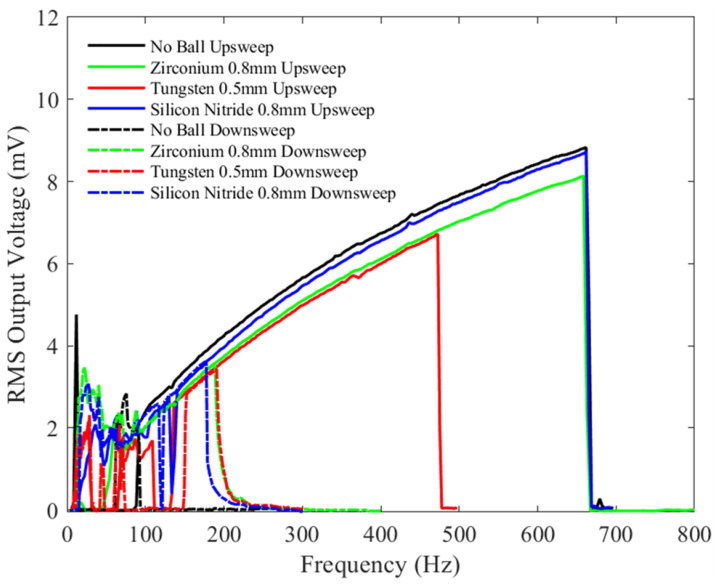
The measured RMS voltage output from the device with different balls under a 3 V bias voltage and an input mechanical vibration of 2 g peak-to-peak acceleration under a variety of frequencies.

**Table 1 sensors-23-05296-t001:** Device parameters.

Parameter	Value
Total device area	11.15mm×11.15mm
Thickness of the silicon layer	25μm
Minimum electrode width	9μm
Maximum electrode width	30μm
Electrodes overlap length	400μm
Electrode length	450μm
Nominal gap between the electrodes (narrowest point)	14μm
Nominal gap between the electrodes (wide point)	31μm
Number of electrode pairs	80
Size of the cavity housing the ball	1mm×1.35mm
Thickness of the parylene film	0.2μm
Length of the silicon shuttle mass	6.8mm
Width of the silicon shuttle mass	5.5mm

**Table 2 sensors-23-05296-t002:** Microball parameters.

Material	Diameter	Mass
Tungsten Carbide (WC)	0.5mm	$1.03\times 10^{-6}\;\mathrm{kg}$
Tungsten Carbide (WC)	0.8mm	$4.22\times 10^{-6}\;\mathrm{kg}$
Zirconium Dioxide $\left({\mathrm{ZrO}}_{2}\right)$	0.8mm	$1.52\times 10^{-6}\;\mathrm{kg}$
Silicon Nitride $\left({\mathrm{Si}}_{3}N_{4}\right)$	0.8mm	$0.86\times 10^{-6}\;\mathrm{kg}$

## Data Availability

Data are contained within the article.

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
