# Peer review of "Broadband Vibration-Based Energy Harvesting for Wireless Sensor Applications Using Frequency Upconversion"

_sensors, 2023, doi:10.3390/s23115296_

Round 1

Reviewer 1 Report

In this paper the authors designed, fabricated, and characterized an electrostatic MEMS energy harvester with a non-uniform comb design to explore their potential for frequency up-conversion. A micro ball was added within a cavity of the shuttle mass which serves as an additional springless mass to increase inertia. The authors also claim that the trapezoidal-shaped electrode design is compatible with the common MEMS production techniques with novel criteria.

However, there are still a few points need to be explained before publishing their work:

1.      Lack of the detailed fabrication process of the devices, which is the main contribution to this work.

2.      The micro-fabrication process was performed in two batches with different yields. The first batch resulted in a 20% yield while the second batch resulted in an 80% yield with an improved process.

What is the difference between the two batches?

3.      ( 80-280 Hz) and wider bandwidths ( 80- 380 Hz)

Are they coming from the same batch?

4.      The range of frequencies where the power is 0.707 of the peak power is 213 known as the half-power or 3-dB bandwidth, which is marked using a blue arrow.

The sentence is not clear, what is the 213?

5.      Figure 7 displays frequency sweeps for device #1(as labeled in Fig. 5), showing the RMS output voltage for upsweeps and downsweeps at three different excitation accelerations (1g, 1.5g, and 2g) with a bias voltage of 4 V.

Why 4V is used here compared with 3 V in Figure 6?

6.      Comparing your device to previously reported device [1] shown in Figure 6(b) and 6(a), respectively, although you have a better BW and RMS output voltage but their device did not suffer from hysteresis between upwnsweeps up and downsweeps.

7.      The results shown in Figs. 12-17 illustrate the RMS output voltage as a function of

frequency for upsweep and downsweep scans with and without a micro ball placed inside the cavity with different peak-to-peak acceleration levels and bias voltages.

To sum it up, with or without a micro ball, the RMS output voltage increases strongly 365

when the bias voltage is increased from 1.5 V (Fig. 12-14) to 3 V (Fig. 15-17), depending on 366

the applied peak-to-peak acceleration.

Figure numbers should be revised.

8.      A comparison between your results and more recent published work should be made.

The manuscript needs proper language revision for grammar, punctuation marks, etc.

Reviewer 2 Report

This paper focus on solving the problem of the limitation of typical electrostatic energy harvesters’ power output and energy conversion. The author proposed an impacted-based electrostatic energy harvester to increase energy output. At the same time, experimental results have shown that the system operates over a relatively wide frequency range, with the lower limit far below the natural frequency of the device.

After I review this article, I think this paper can be accepted if the problems listed below can be solved or add something to them.

1.      In the introduction, you mention three main methods for harvesting energy: electrostatic, piezoelectric, and electromagnetic. However, you do not fully discuss the pros and cons of these approaches and why impact-based electrostatic energy reaper is the best solution you have chosen. Please elaborate further on your reasons for choosing an electrostatic energy reaper and its advantages over other methods.

2.      In describing the experimental equipment and methods, you mention "non-uniform width electrodes" and "spring-free mass," but do not detail how these design features improve energy harvesting efficiency. Please explain in detail how these design elements improve the performance of the equipment and provide the relevant theoretical basis.

3.      In the paper, you mention the variety of experimental equipment used in the study. However, the detailed parameters, manufacturer information and model number of these devices were not provided. Please add this information so that the reader can better understand your experimental setup.

4.      You mentioned that the influence of sphere size and material properties on output power was studied under different conditions, but no specific experimental data and results were given. Please supplement this section and provide detailed data analysis and discussion.

5.      You cite other studies several times in your paper without providing a complete reference. Please supplement the complete list of references in accordance with the relevant format requirements.

As for the writing of the article, I suggest that you proofread the full text carefully to eliminate possible grammatical errors and unclear expressions.

Reviewer 3 Report

This manuscript presents the performance of electrostatic energy harvesters with non-uniform cross-section electrodes and a springless mass. A springless mass was incorporated in an attempt to force collisions under a frequency range. The proposed harvesters can operate over a wide frequency range. However, this manuscript must be significantly improved based on the following comments:

1.-The abstract should be enhanced by considering the main advantages of the proposed device. In addition, the materials and fabrication process of the device should be included. The main results should be added. A conclusion must be written.

2.-The introduction should add the advantages and limitations of energy harvesters published in the technical literature, including different transduction mechanisms such as piezoelectric, electrostatic, and triboelectric. In addition, this section should consider the main advantages and limitations of the proposed energy harvester compared to others reported in the technical literature.

3.-The authors can include a table with the proposed device’s main performance parameters compared with other energy harvesters reported in the literature. What is the output power density of the proposed energy harvester?

4.-Figures 1, 2, 4a, and 7 are equal to Figures 1,2, 3, and 5 of paper J. Li et al., "A Novel Comb Design for Enhanced Power and Bandwidth in Electrostatic MEMS Energy Converters," 2023 IEEE 36th International Conference on Micro Electro Mechanical Systems (MEMS), Munich, Germany, 2023, pp. 728-731, doi: 10.1109/MEMS49605.2023.10052590. Figures 1, 2, 4a, and 7 must be modified or these figures must incorporate the copyright of IEEE.

5.- The section on the device description is weak. This section must be significantly enhanced. It must add more information on the design stage of the proposed device, incorporating better figures or schematic views of the design of the different mechanical and electrical parts of the proposed device. Figures or schematic views on the operating principle of the proposed device must be added. The description of the proposed device’s design and main components must be improved.

6.-This manuscript should include more technical information on the fabrication process of the proposed device.

7.-The authors should consider more discussions on the challenges of the proposed device. For instance, the effect of the residual stress of the beams on the performance of the proposed device. What is the effect of the temperature on the performance of the device?

8.-Discussions on the reliability of the device should be considered. 

9.-What is the future research work?

10.-The conclusions should be improved by incorporating the new modifications of the manuscript.

English language could be improved.

Reviewer 4 Report

This paper reports an experimental study on a nonlinear electrostatic vibration energy harvester with wide bandwidth.

The paper is interesting and well written. It merits consideration for publication, once the following issues are considered by the authors:

1.

This paper reports experiments only. This should be clearly indicated in the title and in the abstract.

The paper is missing any modelling attempt, but addresses some interesting questions, which could be taken into account in a next modelling work.

2.

The paper is missing a clear application field where the proposed device could be potentially adopted. Energy harvesting have been investigated for a variety of applications. A broader view to the potential reader should be given. Some recent applications are provided below as examples in energy harvesting from ocean waves, wind, passing trains, walking persons

- “Integrated study of triboelectric nanogenerator for ocean wave energy harvesting: Performance assessment in realistic sea conditions”, Nano Energy 84 (2021) 105890

- “Research on Piezoelectric Energy harvesting from Multi Direction Wind-Induced Vibrations”, IOP Conf. Series: Earth and Environmental Science 617 (2020) 012014

- Energy harvesting from the vibrations of a passing train: Effect of speed variability, Journal of Physics: Conference Series 744(1),012080, 2016

- “Piezoelectric energy harvesting from human walking using a two stage amplification mechanism”, Energy 189 (2019) 116140.

3.

In other applications, intermitted or time-limited excitations are considered. These cases are treated, for example in

- Challenges for Energy Harvesting Systems Under Intermittent Excitation, IEEE J. Emerging Sel. Top. Circuits Syst., 4(3), pp. 364–374, 2014

- Harvesting Energy From Time-Limited Harmonic Vibrations: Mechanical Considerations, Journal of Vibration and Acoustics, 2017, 139, 051019

I believe that the interested reader should be made aware that there are a series of different input from which to harvest kinetic energy.

4.

Better clarify the contribution of the paper at the end of the introduction. A clear technological contribution is stated to allow commercial fabrication. Is there any scientific contribution as well?

5.

Fig. 1(a) is not clear. It would be better to make it 2D, and align it to the in-plane axes. An indication of the representative scale of the device in millimetres would be useful.

6.

The text in fig 3 is hardly visible. If it is important, it should be enlarged, otherwise should be removed.

7.

The nonlinearity coming from the presented device should be better emphasised and described.

8.

In fig. 5 there is a jump-down phenomenon at 280 and 380 Hz. This is a well-established property of nonlinear devices. However, along with a resonant branch at higher amplitudes, there should also be a non-resonant branch at lower amplitudes. The two responses coexist.

How is the chance of the device to respond at the highest amplitude, compared to the lowest?

It is believed that the actual response depends on the initial conditions. Some more insight is expected on this issue.

9.

The experimental study is very interesting and well reported. However, at the end, a clear summary of the results is missing. It would be interesting to report some concluding remarks in a more compact and comprehensive manner. Perhaps using a table and defining some quantitative metrics.  

10.

This study opens the way to a next modelling paper which addresses the issues reported in the experimental work. A summary of the opening questions which remain unanswered or that involve a modelling insight would be interesting to the potential reader.

Reviewer 5 Report

This article discusses the development of an impacted-based electrostatic energy harvester that aims to address the limitation of typical electrostatic harvesters that do not produce sufficient power output at low frequencies. The device uses electrode collision to trigger frequency up-conversion and generate a secondary high-frequency free oscillation of the electrodes. This design allows for additional energy conversion cycles, which increases energy output. The article provides some useful insights into the limitations of traditional electrostatic harvesters and the potential of impacted-based designs to overcome these limitations.

Overall, the article seems to provide a clear and detailed explanation of the experimental setup and results obtained. However, there are a few comments and improvements that could be made to improve the clarity and impact of this article:

1. The article could benefit from a clear statement of the research question or hypothesis being tested. This would help readers understand the significance of the research and the implications of the results obtained.

2. The article should provide a more detailed description of the device design, including the materials used and the fabrication process. This information would be helpful in understanding the reasons for the observed differences in device response and the potential impact of these differences on commercialization.

3. The article should provide a more detailed explanation of the method used to determine the bandwidth of the devices. This would help readers understand the significance of the results obtained and the potential applications of the device.

4. The article should provide a more detailed discussion of the potential limitations of the study, including the small sample size and the potential impact of variations in the fabrication process.

5. The article should provide a more detailed explanation of the potential applications of the device, including the potential impact on energy harvesting and other fields.

There are also some areas for improvement.

Firstly, the article would benefit from a clearer explanation of the device's operating principle, particularly for readers unfamiliar with electrostatic energy harvesters.

Additionally, it would be useful to include more details about the experimental setup and testing methodology, such as the range of frequencies tested, the amplitude of the input vibrations, and the power output of the device.

Furthermore, the article could benefit from a discussion of the potential applications of impacted-based electrostatic energy harvesters. While the article mentions wireless sensor networks, wearable technology, and environmental and structural monitoring as potential applications, it would be helpful to provide more specific examples of how these devices could be used in practice.

There are a few details missing in Section 3. Experimental setup, such as the values of the input voltage and load resistor used for the electrical characterization, and the specific range of frequencies and acceleration amplitudes used for the mechanical excitation. These parameters are crucial to ensure the reproducibility and reliability of the experimental results.

Additionally, it is not clear whether any environmental conditions such as temperature or humidity were controlled during the experiments, which can affect the device performance. Overall, providing more details on the experimental parameters and environmental conditions would help in better understanding and replicating the experimental results.

表單的頂端

The conclusions could be improved by providing more specific information on the experimental test results, such as the power output and bandwidth achieved with and without the micro ball.

Additionally, it would be helpful to provide a more detailed explanation of the theoretical model that could be developed based on the results of this study, as well as the potential applications of the device in practical settings.

Furthermore, the conclusions could benefit from a discussion of the limitations of the device and the potential for future improvements. For example, the paper mentions that heavier masses may lead to damage to the silicon die or overall device structure, but it does not elaborate on this issue or provide suggestions for overcoming it.

Overall, the article provides some interesting insights into the development of impacted-based electrostatic energy harvesters and their potential to overcome the limitations of traditional designs. However, there is room for improvement in terms of clarity of explanation and additional details about experimental testing and potential applications.

Major revision is required.

Minor editing of English language required.

Round 2

Reviewer 1 Report

Most of the comments were answered.

Minor editing needed

Reviewer 3 Report

The authors have improved their manuscript based on the reviewer's comments.

The English grammar is good.

Reviewer 4 Report

Authors have responded somehow to my comments

small English typos

Reviewer 5 Report

The authors have answered all of my quesions.

English writing is acceptable.